# Therapeutic Single Compounds for Osteoarthritis Treatment

**DOI:** 10.3390/ph14020131

**Published:** 2021-02-06

**Authors:** Hyemi Lee, Xiangyu Zhao, Young-Ok Son, Siyoung Yang

**Affiliations:** 1Department of Biomedical Sciences, Ajou University Graduate School of Medicine, Suwon 16499, Korea; hyemi0320@ajou.ac.kr; 2Department of Pharmacology, Ajou University School of Medicine, Suwon 16499, Korea; 3Degenerative InterDiseases Research Center, Ajou University School of Medicine, Suwon 16499, Korea; 4Department of Animal Biotechnology, Faculty of Biotechnology and Interdisciplinary Graduate Program in Advanced Convergence Technology and Science, Jeju National University, Jeju 63243, Korea; zhaoxiangyu@jejunu.ac.kr; 5Bio-Health Materials Core-Facility Center, Jeju National University, Jeju 63243, Korea; 6Practical Translational Research Center, Jeju National University, Jeju 63243, Korea

**Keywords:** natural compound, Osteoarthritis, treatment

## Abstract

Osteoarthritis (OA) is an age-related degenerative disease for which an effective disease-modifying therapy is not available. Natural compounds derived from plants have been traditionally used in the clinic to treat OA. Over the years, many studies have explored the treatment of OA using natural extracts. Although various active natural extracts with broad application prospects have been discovered, single compounds are more important for clinical trials than total natural extracts. Moreover, although natural extracts exhibit minimal safety issues, the cytotoxicity and function of all single compounds in a total extract remain unclear. Therefore, understanding single compounds with the ability to inhibit catabolic factor expression is essential for developing therapeutic agents for OA. This review describes effective single compounds recently obtained from natural extracts and the possibility of developing therapeutic agents against OA using these compounds.

## 1. Introduction

### 1.1. Background

Osteoarthritis (OA) is a serious chronic degenerative disease of the joints that is common in middle-aged and elderly people. The main clinical manifestations of OA are degeneration of the articular cartilage and changes in the subchondral bone structure. When joint cartilage is completely lost following disruption of cartilage homeostasis through the induction of catabolic factors as well as down-regulation of anabolic factors., the bones and soft tissue structures around the joint are altered, resulting in joint pain, swelling, deformity, and disability [1,2]. Molecular evidence clearly suggests that nuclear factor-κB (NF-κB) signaling plays a central role in matrix metalloproteinase-3 (Mmp3), -13, and cyclooxygenase-2 (COX2) expression and suppresses sex-determining region Y (SRY)-box 9 (Sox9) expression, thus down-regulating the synthesis of cartilage extracellular matrix proteins such as collagen type II alpha-1 (COL2A1) [3,4,5]. In the USA, approximately 10% of adults over the age of 60 years show clinical symptoms of OA [6]. According to the Center for Disease Control and Prevention estimates, 78 million US adults will have arthritis in the year 2040 [7]. As the population ages, the number of patients with OA is expected to increase [8,9]. The overall morbidity of joint diseases in the USA is 15%, of which more than 40% are attributed to OA, showing a prevalence that is proportional to age. The number of females with OA is higher than the number of males, and the incidence of joint diseases in people over 65 years of age is more than 68% [10]. Unfortunately, diagnosing OA by magnetic resonance imaging is difficult until the end of cartilage destruction. The best treatment for OA is to maintain healthy cartilage and administer treatment at an early stage, before the induction of severe osteoarthritic cartilage destruction. Although there is currently no Food and Drug Administration-approved indication for using glucosamine sulfate as supplement to treat OA, many elderly people use natural extracts to protect the joint cartilage from inflammation. Previous reports suggested that *Boswellia serrata*, *Arnica montana*, *Apis mellifera*, *Psoralea corylifolia*, *Rhizome coptidis*, *Betulae cortex*, *Harpagophytum procumbens*, *Phellodendron amurense*, *Symphytum officinalis*, and *Withania somnifera* suppressed proinflammatory cytokines and induced Mmps and Cox2 expression by blocking NF-κB. Interestingly, *Achyranthes bidentata* and *Bauhinia championii* can regulate the cell cycle to protect against osteoarthritic cartilage destruction [11,12]. Although the mechanisms of action of natural extracts are ambiguous because these extracts contain a mixture of various active compounds, previous reports suggested that natural extract for osteoarthritis mainly regulate one pathway such as NF-kB pathway [11,12]. Thus, it is important to determine the functional compounds present in natural extracts and characterize the signaling pathway by each functional compound to develop suitable therapeutic options.

### 1.2. OA Clinical Treatment

Treatment of degenerative arthritis has both prophylactic and therapeutic aspects; the prophylactic aspects have been emphasized in the past, whereas the therapeutic aspects have been highlighted in recent studies [11,12]. At present, clinical treatment of OA is mainly divided into two modes: drug injection and oral administration. Drug injection involves the direct injection of drugs, such as sodium hyaluronate and glucocorticoids, into the knee cavity. Oral administration further includes two categories of drugs: synthetic drugs, which include non-steroidal anti-inflammatory drugs; antipyretics; analgesics; and cartilage-protecting drugs, such as chondroitin sulfate, and different natural compounds, including glycosides, polyphenols, alkaloids, flavonoids, anthraquinones, curcumin, and triptolide. These different drugs have varying pharmacological mechanisms and efficacy in OA treatment.

## 2. Candidate Therapeutic Agents

### 2.1. Mixture Compounds

#### 2.1.1. *Cirsium japonicum* var. *maackii*

*Cirsium japonicum* var. *maackii* (CJM), a member of the family composite, is a perennial herb and medicinal plant listed in the Korea, China, and Japan pharmacopoeias as a traditional antihemorrhagic [13], antihypertensive [14], and uretic medicine [15]. CJM is reported to exert anti-inflammatory and anticancer effects and help prevent diabetic complications and oxidative stress-related diseases. CJM also has neuroprotective effects; in the cells treated with amyloid beta 25-35–treated cells, CJM decreased reactive oxygen species (ROS) accumulation and pro-inflammatory cytokine levels and regulated the expression of apoptotic factors [8]. CJM extracts also show antihepatitis activity. For instance, pretreatment with CJM extract decreased the hepatotoxicity of *tert*-butyl hydroperoxide as well as oxidative damage and increased heme oxygenase 1 (HO-1) and nuclear factor erythroid 2-related factor 2 (NRF-2) expression [16]. CJM comprises several single compounds and well-known compositions including apigenin, cirsimarin, cirsimaritin, hispidulin, and luteolin [17,18,19].

A recent study reported that CJM can block osteoarthritic cartilage destruction [20]. CJM reduces the expression of MMP3, MMP13, ADAMTS4, ADAMTS5, and COX-2 induced by IL-1β, IL-6, IL-17, and tumor necrosis factor (TNF)-α and blocks destabilization of medial meniscus (DMM)-induced cartilage degradation in a mouse model. Further, CJM suppresses activation of hypoxia inducible factor 2 α (HIF-2α) that directly regulates the expression of MMP3, MMP13, ADAMTS4, IL-6, and COX-2. Particularly, in mice subjected to intra-articular injection with adenovirus-HIF-2α, oral administration of CJM resulted in reduced HIF-2α-induced cartilage destruction compared to the control group [17].

#### 2.1.2. Seomae Mugwort

*Artemisia argyi*, a natural herb used in food, tea, and traditional medicine, has antioxidant, anti-inflammatory, and gastroprotective activities. Seomae mugwort (SM), a Korean native variety of *Artemisia argyi*, is a local-specific resource registered by the Korea Forest Service (registration no. 42, 2013, 09. 27). SM contains several compounds, such as volatile chemicals, polyunsaturated fatty acids, phenolic compounds, vitamin C, and essential amino acids as well as high levels of jaceosidin and eupatilin. The anti-inflammatory effects of SM have been demonstrated. A polyphenolic mixture of SM was shown to reduce the activity of macrophages by inhibiting nitric oxide production, inducible nitric oxide synthase and pro-inflammatory cytokine expression, mitogen-activated protein kinase (MAPK) phosphorylation, and nuclear factor kappa-light-chain-enhancer of activated B cells (NF-κB) activity in RAW 264.7 cells treated with lipopolysaccharide [21].

The medical effect of SM in OA has also been analyzed. In an in vitro study, SM suppressed IL-1β-induced MMP3 and MMP13 expression as well as IL-6 and TNF-α expression and decreased the expression of ADAMTS4 and ADAMTS5. Upon oral administration of SM to DMM-induced OA mice, protection against osteoarthritic cartilage degradation was observed; the protective effect of SM on OA was related to suppression of NF-κB and MAPK signaling because SM reduced inhibitor of. κB (IκB) degradation and JNK phosphorylation [21].

#### 2.1.3. *Capparis spinosa* L.

*Capparis spinosa* L. is a vine plant of the genus *Capparis* belonging to the white cauliflower family. It has antibacterial, anti-inflammatory and antioxidant activities. The methanol extracts of its flower buds mainly contain flavonoids, such as kaempferol and quercetin derivatives [22].

Panico et al. [23] showed that methanol extracts of capers (10, 100, and 200 mg/kg) inhibited the release of ROS and PGE_2_ from IL-1β-treated human chondrocytes in a concentration-dependent manner; it also inhibited IL-1β expression in vitro. The release of nitric oxide (NO) from human chondrocytes was increased, but the relationship was not concentration dependent. Further, the extract reversed the decrease in glycosaminoglycan (GAG) release from chondrocytes induced by IL-1β in a concentration-dependent manner. These results indicate that the methanol extract of capers has cartilage-protective effects and can be used to prevent and treat OA. Furthermore, Maresca et al. [24] demonstrated that *C. spinosa* extracts relieved pain related to rheumatoid arthritis and OA after a single administration. They used models of rheumatoid arthritis and OA induced by intra-articular administration of complete Freund′s adjuvant and monosodium iodoacetate (MIA), respectively; different preparations of *C. spinosa* were acutely administered, resulting in significantly reduced hypersensitivity to mechanical noxious stimuli as well as spontaneous pain evaluated as hind limbs bearing alterations in both models.

### 2.2. Single Compounds

#### 2.2.1. Flavonoids

Flavonoids are a group of phytonutrients found in most fruits and vegetables and function as plant secondary metabolites. Flavonoids are well-known pigments present in most angiosperm families and are present in all parts of the plants (Figure 1). These compounds typically have a 15-carbon skeleton with a C6-C3-C6 structure including two phenyl rings and a heterocyclic ring. Depending on the linkage between the phenyl ring and heterocyclic ring as well as the degree of oxidation and unsaturation of the heterocyclic ring, flavonoids are classified as flavones, flavonols, isoflavones, chalcones, and anthocyanins [25]. Many studies showed that flavonoids have antibacterial, anticancer, antiviral, antiallergic, and anti-inflammatory effects [26,27,28,29], with some flavonoids reported to aid osteoarthritis. Here, we introduce flavonoids which have positive effects on osteoarthritis.

##### Apigenin

Apigenin, i.e., 4′,5,7-trihydroxyflavone, is a flavonoid abundant in *C. japonicum* var. *maackii*, chamomile, parsley, celery, basil, and oregano [25,26,27,28]. Apigenin is reported to exert antioxidant, anti-inflammatory, and antiapoptotic effects and suppress cancer development. Apigenin also acts as an antidiabetic compound. In some studies, apigenin was reported to improve renal dysfunction by decreasing the levels of transforming growth factor-β1, type IV collagen, and fibronectin and reduce blood glucose levels by stimulating glucose-induced insulin secretion [29,30].

Apigenin is also effective for treating OA. In an in vitro study, apigenin suppressed the expression and proteolytic activity of MMP3 in rabbit articular chondrocytes and rabbit knee joints; additionally, apigenin decreased the expression of MMP1, MMP3, MMP13, a disintegrin and metalloproteinase with thrombospondin motifs (ADAMTS)-4, and -5 in rabbit articular chondrocytes treated with interleukin (IL)-1β [31]. In an in vivo study, apigenin attenuated cartilage erosion, bone loss, and catabolic factors and reduced the expression of pro-inflammatory cytokines in an MIA-induced rat OA model [32]. The mechanism of action of apigenin was further identified by Cho et al. [20], who reported that apigenin blocked hypoxia-inducible factor (HIF)-2α-induced osteoarthritic cartilage destruction, downregulated HIF-2α, MMP3, MMP13, ADAMTS4, IL-6, and COX-2 expression, and suppressed HIF-2α-induced MMP3, MMP13 and COX-2 expression by regulating HIF-2α expression (Figure 2). This is because apigenin modulates the NF-κB and JNK signaling pathways required for HIF-2α regulation. This study defined the inhibition mechanisms of apigenin against OA in vitro and in vivo; however, this study did not determine the upstream molecules of NF-κB and JNK signaling. HIF-2α is a key regulator in OA, and HIF-2α expression induces the expression of catabolic factors that induce OA by degrading cartilage and inducing inflammation. Therefore, apigenin may be among the first drugs targeting HIF-2α against OA.

##### Cirsimarin and Cirsimaritin

Cirsimarin and cirsimaritin are flavonoids present in *C. japonicum* var. *maackii*. Cirs imaritin is 5,4′-dihydroxy-6,7-dimethoxyflavone and cirsimarin is cirsimaritin 4′-*O*-glucoside [33,34]. Cirsimarin is reported to exert an anti-lipogenic effect by reducing intra-abdominal fat accretion [35]. In lipopolysaccharides-stimulated RAW 264.7 cells, cirsimarin inactivated the Janus kinase/signal transducer as well as the activator of transcription and interferon regulatory transcription factor-3 signaling pathways and decreased the expression of inducible nitric oxide synthase (iNOS) and COX2 as well as that of pro-inflammatory cytokines, such as IL-6 and TNF-α, indicating that cirsimarin has anti-inflammatory activity [18]. Further, cirsimarin has antilipolytic, antiproteinuric, and antioxidant activities. Cirsimaritin also has anti-diabetes effects; cirsimaritin has been reported to reduce caspase-8 and 3 activities and increase B-cell lymphoma 2 (BCL-2) expression [36]. Cirsimaritin exerts anti-lung cancer effects on NCIH-520 cells by inhibiting proliferation compared to other cell lines and increasing the levels of apoptosis and ROS [37]. Cirsimaritin can also suppress influenza A virus replication by suppressing the NF-κB/p65, c-Jun N-terminal kinase (JNK), and p38 signaling pathways but does not change viral attachment and release. In line with its anti-inflammatory effects, cirsimaritin downregulates the expression of pro-inflammatory cytokines, such as TNF-α, IL-1β, IL-8, and IL-10, as well as COX2 [38]. The in vivo effects of cirsimarin or cirsimaritin on OA have not been reported, although these compounds were shown to have anti-inflammatory effects (Figure 2). Cho et al. performed in vitro analysis and showed that cirsimarin and cirsimaritin decreased IL-1β-induced HIF-2α and COX2 expression but not MMP3 and MMP13 expression [20].

##### Jaceosidin

Jaceosidin [5,7-dihydroxy-2-(4-hydroxy-3-methoxyphenyl)-6-methoxy-4*H*-chromen-4-one], a major component of SM, is a pharmacologically active flavonoid [39]. Jaceosidin reduces the oxidation of low-density lipoprotein, which is related to the inflammatory process of atherosclerosis [40]. Jaceosidin also exerts anti-inflammatory effects on lung injury by decreasing TNF-α, IL-6, and IL-1β expression and increasing IL-4 and IL-10 expression in bronchoalveolar lavage fluid [41]. Further, when *db*/*db* diabetic mice were orally administered jaceosidin, their fasting blood glucose levels and insulin resistance were decreased through upregulation of the insulin receptor pathway. Administration of jaceosidin also attenuated the accumulation of glycation end products, decreased vascular endothelial growth factor-alpha protein levels in the kidney, and increased copper- and zinc-superoxide dismutase activity. Overall, jaceosidin has healing effects on diabetic nephropathy [42].

Similar to SM, jaceosidin shows protective effects against OA. Jaceosidin suppresses IL-1β-induced activation of NF-κB signaling and decreases IL-1β-, IL-6-, and TNF-α-induced expression of MMP3 and MMP13 and ADAMTS4 and ADAMTS5 expression [21]. In DMM-induced OA mice, jaceosidin attenuated cartilage destruction; particularly, this protection was more effective for combined treatment with jaceosidin and eupatilin (Figure 3) [21]. As only Yang et al. have reported the effects of jaceosidin on OA, further studies are needed to confirm this result.

##### Eupatilin

Eupatilin [2-(3,4-dimethoxyphenyl)-5,7-dihydroxy-6-methox-ychromen-4-one] is a pharmacologically active flavone and a type of flavonoid which has anti-cancer, antioxidant, and anti-inflammatory effects. Eupatilin can suppress renal cancer growth by inhibiting the expression of miR-21, which targets yes-associated protein 1 [43]. Regarding its anti-inflammatory effect, eupatilin downregulated the expression of pro-inflammatory cytokines, such as IL-1β, TNF-α, and IL-6, in an anaphylactic shock model by regulating NF-κB and MAPK signaling [44]. 

In some studies, eupatilin showed an inhibitory effect on OA. For instance, paw withdrawal latency and threshold were increased when eupatilin was orally administered to experimental OA model rats that had been injected MIA [45]. Eupatilin treatment reduced the Mankin score (an OA disease grade), attenuated cartilage degradation, and decreased the number of osteoclasts in the subchondral bone of an OA rat model [45]. Eupatilin also downregulated the expression of MMP-13, IL-1β, IL-6, iNOS, and nitrotyrosine in an OA rat model compared to in the control [45]. iNOS synthesizes NO to contribute to upregulating the expression of catabolic factors. The effects of eupatilin effects were also investigated in vitro; eupatilin suppressed IL-1β-induced expression of MMP3, MMP13, ADAMTS5, and tissue inhibitor of metalloproteinase-1 in human chondrocytes by reducing JNK phosphorylation (Figure 3) [45]. In another study, eupatilin blocked the apoptosis of chondrocytes stimulated with IL-1β by activating autophagy through upregulation of sestrin2 expression and downregulation of mechanistic target of rapamycin phosphorylation [46]. However, Jeong et al. study used only an MIA OA model, which is not suitable for evaluating mild progression of OA. In addition, Lou et al. did not evaluate autophagy flux; there is limitation information on whether eupatilin-induced autophagy has positive or negative effects on OA.

##### Genistein

Genistein is an isoflavone compound extracted from soybeans reported to exert anti-cancer effects; it decreases HIF-1α levels in breast cancer cells by binding with the factor inhibiting HIF site on HIF-1α [47]. Genistein is also reported to be effective against obesity during menopause, as its administration suppresses estrogen-deficiency-induced obesity and hepatic lipogenesis by reducing NF-κB activity in an ovariectomized and high-fat diet rat model [48]. 

Genistein was also reported to suppress IL-1β-induced apoptosis and increase collagen II and aggrecan expression in human chondrocytes cell line, CHON-001 [49]. Through flow cytometry analysis and enzyme-linked immunosorbent assay, genistein was found to decrease chondrocyte apoptosis and TNF-α levels, respectively [49]. However, this study did not define the apoptosis mechanisms, e.g., caspase dependent or independent. When genistein was orally administrated in an anterior cruciate ligament transection rat model, the collagen and acid glycosaminoglycan content was increased compared to that in the control OA rat model, whereas the level of TNF-α and IL-1β decreased [49]. Additionally, genistein showed anti-inflammatory effects in IL-1β-stimulated human OA chondrocytes [50]. Genistein attenuated NOS2, COX2, and MMP expression in IL-1β-stimulated chondrocytes through activated nuclear factor erythroid 2-related factor 2 (NRF2)/heme oxygenase-1 (HO-1) signaling. Overall, genistein may be useful as a potential treatment for postmenopausal OA. Genistein, an estrogen mimic, is a phytoestrogen used to reduce the adverse effects of estrogen and binds to estrogen receptors [51]. Genistein increased the effects of insulin, resulting in enhanced sulfate-uptake by female bovine articular chondrocytes in the absence of 17β-estradiol [52].

##### Epigallocatechin Gallate (EGCG)

Green tea has high medicinal value and a long history of clinical application. Green tea is rich in polyphenolic compounds, which mainly have anti-inflammatory, antioxidant, and antitumor effects. Epigallo-catechin-3 gallate (EGCG) is the main active ingredient isolated from tea polyphenols, and its antioxidant effect has shown protective effects on erythrocytes. Insertion of EGCG into the outer monolayer of erythrocytes inhibited the access and veneniferous effect of oxidant molecules into erythrocytes [53]. Furthermore, EGCG can inhibit NO release and iNOS expression in human chondrocytes stimulated with IL-1β in vitro; it may target the NF-κB signaling pathway by inhibiting degradation of the NF-κB inhibitor, IκBα, in the cytoplasm, thereby inhibiting NF-κB transport to the nucleus [54]. EGCG can also inhibit the mRNA and protein expression of MMP-1, MMP-13, and COX-2 in human or mouse chondrocytes under IL-1β stimulation, thereby inhibiting GAG release from cartilage tissues [55,56]. Furthermore, numerous studies suggested that EGCG can be used as a food dietary supplement for arthritis [57,58]

##### Butein

Butein (2′,3,4,4′-tetrahydroxy chalcone), a polyphenolic compound isolated from the stem bark of cashews and *Rhus verniciflua* Stokes, has been reported to have various biological activities, including antioxidant, antifibrosis, anti-inflammatory, and antitumor activities [59].

Butein significantly inhibited IL-1β-induced PGE_2_, COX-2, iNOS, TNF-α, IL-6, and MMP-13 expression [59,60]. Additionally, butein inhibited the mRNA or protein levels of MMP-1, MMP-3, ADAMTS-4, and ADAMTS-5 in IL-1β-exposed chondrocyte [59]. In vivo, butein-treated mice exhibited less Safranin O loss and cartilage erosion as well as reduced subchondral bone plate thickness and synovitis [59]. The mechanisms of this phenomenon involved NF-κB signaling, e.g., IκB-α degradation and NF-κB p65 activation. However, the effects of butein on rheumatoid arthritis have not been examined.

##### Wogonin

The root of *Scutellaria baicalensis* Georgi, a labial plant, contains flavonoids (baicalin, wogonin, wogonoside, and wogonin) and is an active component of *S. baicalensis*. This plant also contains small amounts of sterols and amino acids [61]. *Scutellaria baicalensis* exhibits anti-inflammatory, immune-promoting, and sedative antipyretic properties. Additionally, it has shown antimicrobial, antiallergic, hypotensive, diuretic, hypolipidemic, antiplatelet aggregation and anticoagulant, hepatoprotective, and nephroprotective effects [62].

Wogonin inhibits ROS production and the suppression of catabolic markers including IL-6, COX-2, iNOS, MMP-3, MMP-9, MMP-13 and ADAMTS-4 as well as s-GAG release from IL-1β-treated OA chondrocytes. The inhibitory effect of wogonin was mediated through the suppression of c-Fos/AP-1 activity at the transcriptional and post-transcriptional levels in OA chondrocytes [63]. 

##### Morin

Moraxanthin (Morin) is a natural polyphenol originally isolated from members of the Moraceae family. In vitro and in vivo studies showed that morin has very low toxicity and is well-tolerated for long-term administration [64]. Qu et al. revealed that morin effectively inhibited IL-1β-induced NF-κB activation and decreased the production of NO, PGE_2_, MMP1, and MMP3. In addition, Nrf2 and HO-1 were increased by morin, and knockdown of Nrf2 prevented its anti-inflammatory effects [65]. These effects have only been demonstrated in vitro. 

Another group obtained the same results for morin against IL-1β-exposed rat primary chondrocytes. Treatment with morin attenuated IL-1β-induced proteoglycan loss in the articular cartilage by suppressing catabolic factors, such as MMPs, inflammatory mediators, and pro-inflammatory cytokines [66]. Morin inhibited IL-1β-induced phosphorylation of extracellular signal-regulated kinase and p38 in rat chondrocytes [67]. In an in vivo rat OA model induced by anterior cruciate ligament transection, orally administered morin suppressed cartilage degradation [67].

##### Quercetin

Quercetin is a polyhydroxyflavonoid compound with a chemical name of 3,3′,4′,5,7-pentahydroxyflavonoids. Quercetin is widely present in the flowers, leaves, and fruits of plants. Its pharmacological effects have been extensively studied, including its antioxidant, anticancer, anti-inflammatory, antibacterial, and antiviral activities [68,69].

Oral administration of quercetin with glucosamine or chondroitin significantly improved knee OA pain symptoms [70]. Quercetin can significantly reduce inflammatory mediators, such as IL-1β, TNF-α, IL-6, and other cytokines [69]. Quercetin regulates the expression of MMP-13 and has potential medicinal value [71]. Combined quercetin with palmitoylethanolamide (palmitoylethanolamide) treatment in an OA rat model resulted in low levels of IL-1β and TNF-α and improvement the pain index and histological scores [72]. In a type II collagen-immunized arthritis mouse model, quercetin alone (30 mg/kg) significantly reduced the expression of IL-1β, TNF-α, and IL-6 compared to in the control group [73]. Quercetin is a potential therapeutic drug against OA, targeting TNF-α, IL-1β, and IL-6 [73]. Further, the protective effect of quercetin against OA was investigated in *tert*-butyl hydroperoxide-stimulated rat chondrocytes and DMM rat OA model. Quercetin treatment attenuated oxidative stress and endoplasmic reticulum stress through the sirtuin1/adenosine monophosphate-activated protein kinase signaling pathway [74].

#### 2.2.2. Glycosides

Glycosides contain a glycosidic bond formed by binding of sugar with another functional group to generate an O-, N-, or C-glycosidic bond. Glycosides include some flavonoids and secondary metabolites in plants (Figure 1). The bioactivity of many glycosides has been demonstrated, and traditional use of these compounds has been reported [75,76,77]. Additionally, the effect of glycosides against osteoarthritis were reported, with representative glycosides described below.

##### Paeoniflorin

*Paeonia lactiflora* Pallas is a traditional Chinese natural plant medicine that has been used as an analgesic and anti-inflammatory agent and to improve the immune system for thousands of years in China. The therapeutic effects of *P. lactiflora* Pallas have been listed in the Chinese Experience Medicine books *Treatise on Cold Pathogenic* and *Synopsis of Golden Chamber* [78,79]. Total glycoside of paeony (TGP) is extracted from the roots of *P. lactiflora* Pallas. TGP includes effective components, such as paeoniflorin, hydroxyl-paeoniflorin, paeonin, albiflorin, and benzoyl-paeoniflorin [80]. Paeoniflorin, [(1*R*,2*S*,3*R*,5*R*,6*R*,8*S*)-6-hydroxy-8-methyl-3-[(2*S*,3*R*,4*S*,5*S*,6*R*)-3,4,5-trihydroxy-6-(hydroxymethyl)oxan-2-yl]oxy-9,10-dioxatetracyclo[4.3.1.02,5.03,8]decan-2-yl]methyl benzoate is a monoterpene glucoside, is a major active component of TGP. Paeoniflorin accounts for more than or equal to 40% of TGP.

A study found that paeoniflorin possessed extensive anti-inflammatory immunoregulatory effects. Paeoniflorin can diminish pain, joint swelling, synovial hypertrophy, bone erosion, and cartilage degradation in collagen induced arthritis [81,82]. Further, paeoniflorin suppressed the migration of fibroblast-like synoviocytes from patients with RA or OA patients through blocking the CXCR4-Gβγ-PI3K/AKT signaling [83]. Particularly, some studies reported that paeoniflorin suppresses OA. IL-1β-induced MMP1, MMP3, and MMP13 expression was reduced and issue inhibitor of metalloproteinase-1 expression was increased in chondrocytes pretreated with paeoniflorin [84,85]. Furthermore, paeoniflorin suppressed NF-κB signaling, as indicated by increased inhibitor of NF-κB (IκB) and decreased p65 protein levels [84,85]. Another study reported that paeoniflorin blocked chondrocyte apoptosis induced by IL-1β by downregulating Bcl2 and Bcl-2-associated X-protein levels as well as caspase 3 activity; paeoniflorin also regulated protein kinase B signaling by increasing phosphorylation of this protein [86]. The mechanisms of paeoniflorin on OA have only been demonstrated in vitro; therefore, in vivo animal studies are needed to clarify the mechanisms of the effects of paeoniflorin.

##### Clematis Saponins

Total Clematis saponin extract from *Clematis Florida* Thunb. This plant belongs to the family Ranunculaceae, which includes a wide range of species with a large distribution. Its chemical composition is relatively complex and includes triterpenoid saponins, flavones, and lignans as the main active components. Clematis was found to have anti-inflammatory and analgesic, antitumor, immunosuppressive, and antioxidant effects.

The main mechanism of clematis saponins against OA is inhibition of chondrocyte apoptosis. Total clematis saponin extract (300 µg/mL) can significantly inhibit staurosporin-induced apoptosis of rat cartilage cell lines (RCJ3.1 C.18). It prevents downregulation of the expression of the intracellular antiapoptotic proteins, Bcl-xL and Bcl-2, induced by staurosporin, and inhibits upregulation of the expression of the intracellular proapoptotic protein Bcl-2-associated X-protein induced by staurosporin [87]. Further studies suggest that total clematis saponins target chondrocyte apoptosis through the antiapoptotic protein 14-3-3. The 14-3-3 protein can combine with proteins, such as Bad, to play an antiapoptotic role. Total clematis saponins can inhibit the decrease in 14-3-3 protein expression caused by staurosporin [88]. The total saponins of *Tripterygium* can also inhibit the reduction in 14-3-3 protein expression with phospho-Ser112-Bad and Bcl-xL with the phospho-Ser155-Bad induced by staurosporin [88]. A study of *Clematis chinensis* Osbeck roots showed that treatment with the saponin-rich fraction of *C. chinensis* Osbeck suppressed apoptosis, depolarization of the mitochondrial membrane, and caspase-3 activity in rabbit chondrocytes exposed to sodium nitroprusside, a NO donor [89]. As inhibition of chondrocyte apoptosis is the main mechanism of clematis saponins against OA, other mechanisms should be investigated for drug development.

#### 2.2.3. Non-Flavonoid Polyphenolics

Polyphenols, which are compounds with a polyphenol structure, comprise two subgroups: flavonoids and non-flavonoids (Figure 1). There are three subgroups of non-flavonoids with different structures: phenolic acids, stilbenes, and lignans [90]. Phenolic acids are hydroxybenzoates that contain some phenolic rings and at least one carboxylic acid. Stilbenes commonly have a C6–C2–C6 structure; resveratrol is the most common stilbene. Lignans are low-molecular weight polyphenols that exist as phenolic dimers with a 2,3-dibenxylbutane structure.

##### Resveratrol

Resveratrol, a non-flavonoid polyphenolic compound, was first obtained from the root of *Veratrum grandiflorum* in 1904. Accumulating evidence has shown that resveratrol has anti-inflammatory, antioxidant, immunomodulatory, and antitumor activities.

Resveratrol significantly inhibited the increased clinical scores in rats with OA. Resveratrol suppressed TNF-α, IL-1β, IL-6, and IL-18 expression and decreased caspase-3/9 activity in rats with OA [91]. iNOS, NF-κB, phosphorylated-p-AMP-activated protein kinase, and sirtuin-1 protein expression was significantly suppressed, whereas HO-1 and Nrf-2 protein expression was stimulated in OA rats treated with resveratrol [91]. These results indicate that resveratrol ameliorates inflammatory damage and protects against OA in a rat model through NF-κB and HO-1/Nrf-2 signaling. This study did not show histology images, although animal experiments were performed. The anti-osteoarthritic effects of resveratrol via NF-κB signaling were recently reported. For example, resveratrol regulates NF-κB signaling in IL-1β-induced chondrocyte injury [92]. The anti-inflammatory effects of resveratrol are also involved in NF-κB signaling [93,94] Another study showed that resveratrol can inhibit chondrocyte mitochondrial degradation and apoptosis caused by IL-1β and that its antiapoptosis effect may be mediated through inhibition of caspase-3 expression, cleavage of the DNA repair enzyme PARP, and upregulation of ROS levels induced by IL-1β [95]. Resveratrol can also induce p53 protein degradation in a concentration-dependent manner and inhibit chondrocyte apoptosis caused by p53. Resveratrol may thus exert anti-OA effects by inhibiting chondrocyte apoptosis, and may target multiple proteins in the chondrocyte apoptosis pathway [95]. Only the in vitro mechanism was determined, which is a limitation of this study. Importantly, intra-articular injection of resveratrol exerted a curative effect by preventing inflammation and cartilage destruction [96].

##### Curcumin

Curcumin is the main active ingredient of turmeric (*Curcuma longa* L.) and has been reported to exert anti-inflammatory, antioxidant, lipid-regulatory, antiviral, anti-infective, antitumor, anticoagulation, anti-liver fibrosis, and anti-atherosclerotic effects. It also demonstrates anti-cirrhosis activity, low toxicity, and limited adverse reactions. Curcumin is currently one of the highest-selling natural food colors in the world. It is a food additive approved by the World Health Organization, US Food and Drug Administration, and other organizations in many countries [97].

Many studies previously reported the effect of curcumin on OA. In a mouse model of OA, oral administration of curcumin delayed disease progression and decreased the transcriptional level of catabolic factors, such as MMPs, ADAMTS5, IL-1β, and TNFα, in chondrocytes [98]. In addition, topically applied curcumin nanoparticles were localized within the infrapatellar fat pad and effectively reduced the expression of pro-inflammatory mediators [98]. This study provides the first evidence that single compounds can reach the articular cartilage area and exert therapeutic effects. A study using an MIA-induced OA rat model, with intraperitoneal injection of curcumin, demonstrated decreased levels of IL-6, IL-1β, and TNFα in synovial fluid compared to in a control OA rat model [99]. Curcumin treatment also significantly reduced the level of MyD88 and IκB phosphorylation in the NF-κB signaling pathway [99]. Additionally, it has been reported that dietary supplementation or ointment application reduced osteoarthritic pain in a mouse model and clinical trials [100,101].

#### 2.2.4. Other Compounds

In addition to those mentioned above, many other natural compounds are found in plants and animals and have been used for drug discovery or bioassays because of their bioactivity. As natural compounds, secondary metabolites are used because they confer advantages but are not necessary for survival. Among the compounds not included in the previous category, some compounds reported as effective in osteoarthritis are introduced below.

##### 3′-Sialyllactose

Human milk contains up to 23 g/L oligosaccharides (Figure 1). These oligosaccharides consist of a lactose core elongated with various carbohydrates, such as glucose (Glc), galactose (Gal), fucose (Fuc), *N*-acetylglucosamine (GlcNAc), and *N*-acetylneuraminic acid (Neu5Ac). Neu5Ac, named as sialic acid (SL), is the main monosaccharide representing variants of a 9-carbon carboxylated backbone and 2,3-sialyllactose (3′-SL), which are mainly produced by sialyltransferase (ST3GAL). In a CIA model of rheumatoid arthritis in mice, 3′-sialyllactose decreased paw swelling, the clinical score, pro-inflammatory cytokine levels in the serum, autoantibody production, synovitis and pannus formation, and cartilage destruction. Its protective effect was related to NF-κB signaling by regulating p65 phosphorylation [102]. In atopic dermatitis, 3′-sialyllactose prevents skin inflammation. Administration of 3′-sialyllactose reduced ear, epidermal, and dermal thickness in a mouse model of atopic dermatitis, downregulated the expression of pro-inflammatory and atopic dermatitis-related cytokines, and suppressed mast cell infiltration and IgE levels. Further, 3′-sialyllactose induced transforming growth factor-β-mediated Treg differentiation in an in vitro system [103]. Experiments in both in vitro and ex vivo systems showed that 3′-sialyllactose restored IL-1β-induced reduction of COL2A1 expression as well as promoted the accumulation of sulfated proteoglycan, which is a critical factor for cartilage regeneration in OA development [104]. In addition, 3′-sialyllactose reduced IL-1β-induced up-regulation of MMP3, MMP13, and COX2 expression and showed similar effects in OA-mimicking conditions induced by IL-6, IL-17, and TNF-α [104]. IL-1β-induced reduction of SOX9 expression, a master transcription factor for COL2A1 expression, was also restored by 3′-sialyllactose, supporting previous data that 3′-sialyllactose increased the expression of COL2A1 [104]. 3′-Sialyllactose inhibited MAPK and NF-κB signaling by attenuating extracellular-signal-regulated kinase phosphorylation and IκB degradation, resulting in decreased MMP and COX2 expression [104]. The effect of 3′-sialyllactose was confirmed in a DMM-induced OA mouse model, demonstrating inhibition of cartilage degradation (Figure 4) [104]. Studies of the effects of 3′-sialyllactose studies against rheumatoid arthritis or OA rheumatoid arthritis have been performed using suitable in vitro and in vivo mouse models, although other carbohydrate-type compounds must be evaluated.

##### Triptolide (*Tripterygium wilfordii* Hook. f.)

Triptolide is a diterpene lactone compound containing three epoxy groups isolated from *Tripterygium wilfordii* Hook. f (Figure 1). Triptolide has anti-inflammatory, immunosuppressive, and other activities, and has been widely used in the clinic. Recently, triptolide was reported to exert anticancer effects; intravenous administration of triptolide in mice along with H538 lung cancer cells through the tail vein led to altered microRNA expression related to cell movement along with decreased cell migration and invasion [105]. Triptolide also decreased pro-inflammatory cytokine-induced MMP3 and MMP13 expression in primary rat, bovine, and human chondrocytes as well as in human chondrosarcoma cells and synovial fibroblasts [106,107]. Furthermore, triptolide suppressed OA progression in a DMM-mouse model and inhibited the expression of pro-inflammatory cytokines, which was related to downregulation of the expression of has-miR-20b, which targets the NLR family pyrin domain containing 3 (NLRP3) gene. Triptolide downregulated has-miR-20b expression, leading to increased and decreased NLRP3 and caspase-1 expression, respectively [108]. The exact molecular mechanisms of the effects of triptolide on OA remain unclear.

##### Hyaluronic Acid (HA)

HA, also known as hyaluronan, is a non-sulfated glycosaminoglycan expressed in connective, epithelial, and neural tissues. HA is used to treat intra-articular injection and alleviates pain in OA. The safety of intra-articular HA injections for OA is well-established; however, the most common adverse effect of HA is a self-limited reaction at the injection site [109]. According to a recent study, HA injection is more effective for osteoarthritic pain in the long-term compared to corticosteroids. Further, the effects of HA and corticosteroids in knee function improvement were shown to be similar [110]. Recently, to increase the efficacy of pain relief by HA, combination approaches have been attempted using a PPAR-δ agonist or mesenchymal stem cells [111,112].

## 3. Conclusions

Based on the results of recent research, several active components from plants and natural compounds are effective for treating OA (Figure 5). However, these studies have many limitations.

For example, most studies were limited to in vitro experiments, lacking sufficient animal and clinical experimental data. Additionally, preliminary studies were conducted to determine the anti-OA and cartilage protective effects, with a lack of in-depth discussion on the mechanism of action. Insufficient attention has been given to the toxicity and side effects of the drugs. Moreover, the etiology and pathological mechanisms of OA are unclear, greatly increasing the difficulties of drug research and making the research targets considerably scattered.

## Figures and Tables

**Figure 1 pharmaceuticals-14-00131-f001:**
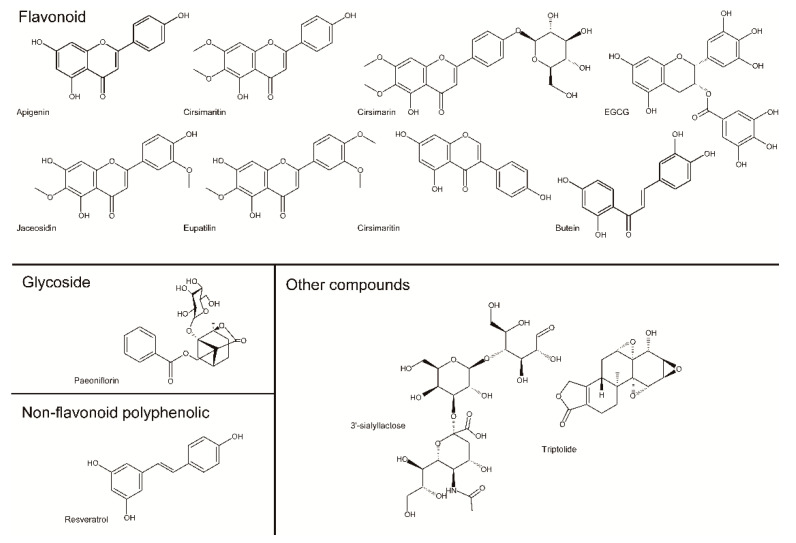
Structures of single compounds.

**Figure 2 pharmaceuticals-14-00131-f002:**
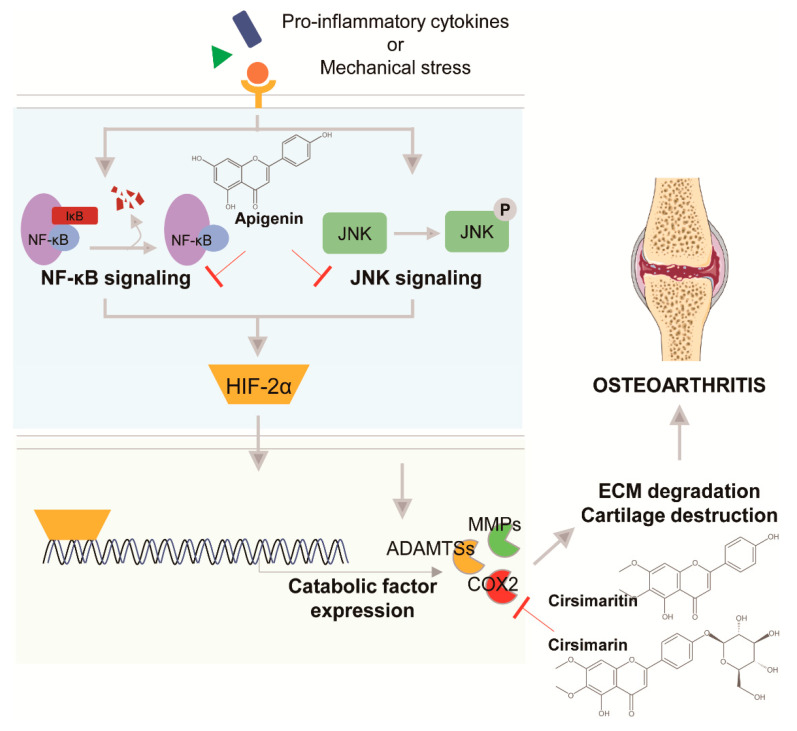
Mechanism of action of apigenin, cirsimarin, and cirsimaritin.

**Figure 3 pharmaceuticals-14-00131-f003:**
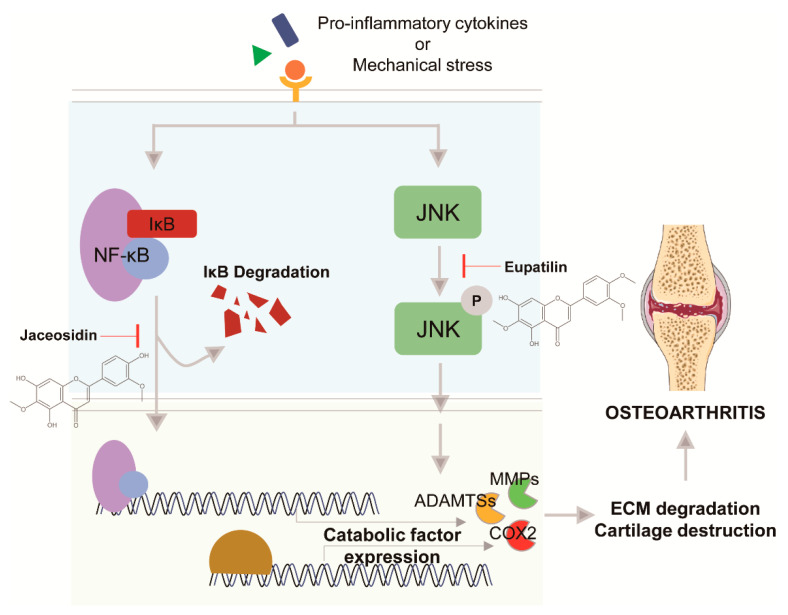
Mechanism of action of jaceosidin and eupatilin.

**Figure 4 pharmaceuticals-14-00131-f004:**
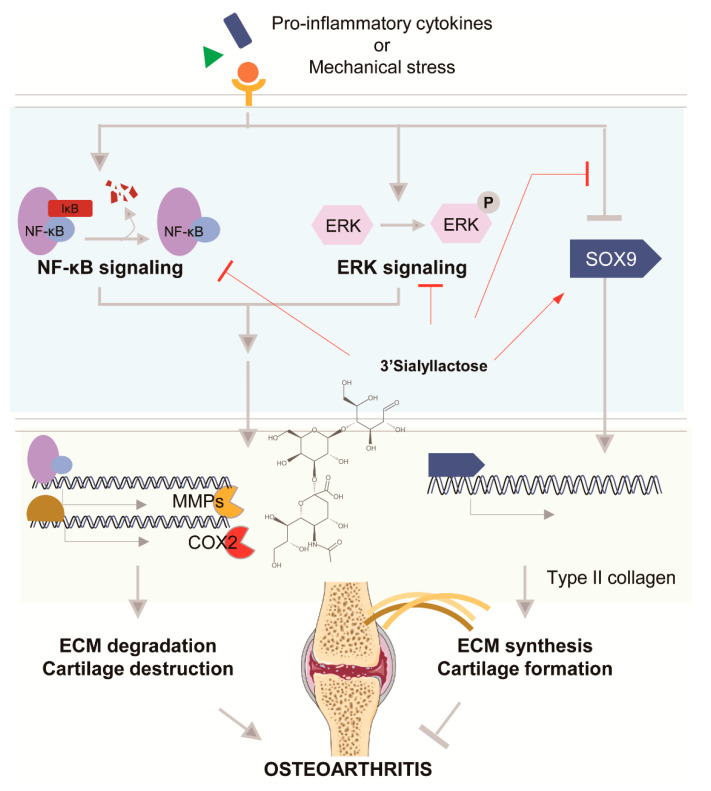
Structure of 3′-sialyllactose.

**Figure 5 pharmaceuticals-14-00131-f005:**
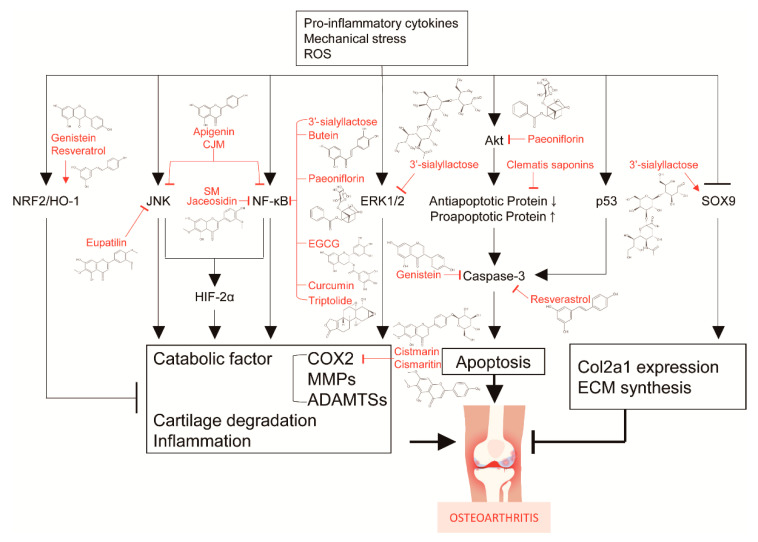
Mechanisms of action of single compounds against osteoarthritis.

## Data Availability

Data is contained within the article.

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
