# Peer review of "Therapeutic Single Compounds for Osteoarthritis Treatment"

_pharmaceuticals, 2021, doi:10.3390/ph14020131_

Round 1

Reviewer 1 Report

It is a well written and comprehensive review by Lee and coworkers about the therapeutic single compounds for OA treatment. I recommend for publication in Pharmacetuicals after the following points are addressed.

1) There are plenty of places the authors should add references to support their claims. For example, references related to the composition of SM should be added (Line 92-94).

2) The authors should add one paragraph to describe 2.2.1 Flavonoids (line 127), which is also true for 2.2.2 Glycosides (line 245), 2.2.4 Non-flavonoid Polyphenolics (line 293), and 2.2. 5 Other Compounds (line 346).

3) Following the comment 2, where is the section 2.2.3?

4) Figure 1, 2, 3, 5, 6, 8, 9, 10, 11, and 13 (for the chemical structures of the compounds) should be combined into one figure, schematically shown all the compounds.

5) The image resolution of figure 4, 7, 12, and 14 is not high enough for publication.

6) Format issues. For example, ‘2.2.5.1.3'-. Sialyllactose’ (line 347).

Author Response

Reviewer #1:

Comments to the Author
Major corrections:

Comment:

[1] There are plenty of places the authors should add references to support their claims. For example, references related to the composition of SM should be added (Line 96-98).

Author response and action:

We agree with the reviewer’s comment and we have added references to increase the credibility of our review. We have corrected the main text as recommended, which has been highlighted in the revised manuscript.

CJM comprises several single compounds with well-known compositions including apigenin, cirsimarin, cirsimaritin, hispidulin, and luteolin [14-16]. (Lines 97–98)

Comment:

[2] The authors should add one paragraph to describe 2.2.1 Flavonoids (line 127), which is also true for 2.2.2 Glycosides (line 245), 2.2.4 Non-flavonoid Polyphenolics (line 293), and 2.2. 5 Other Compounds (line 346).

Author response and action:

Thank you for your comment. We have added a paragraph describing the definition, structure, and feature of each group.

We have added the following information on page 4: “Flavonoids are a group of phytonutrients found in most fruits and vegetables and function as plant secondary metabolites. Flavonoids are well-known pigments present in most angiosperm families and are present in all parts of the plants. These compounds typically have a 15-carbon skeleton with a C6-C3-C6 structure including two phenyl rings and a heterocyclic ring. Depending on the linkage between the phenyl ring and heterocyclic ring as well as the degree of oxidation and unsaturation of the heterocyclic ring, flavonoids are classified as flavones, flavonols, isoflavones, chalcones, and anthocyanins [25]. Many studies showed that flavonoids have antibacterial, anticancer, antiviral, antiallergic, and anti-inflammatory effects [26-29], with some flavonoids reported to affect osteoarthritis. Here, we introduce flavonoids which have effects on osteoarthritis.”

We have also added the structures of the single compounds to Figure 1.

Moreover, based on your comments, we have added the following information to page 10: “Glycosides contain a glycosidic bond formed by binding ofsugar with another functional group to generate an O-, N-, or C-glycosidic bond. Glycosides include some flavonoids and secondary metabolites in plants. The bioactivity of many glycosides has been demonstrated, and traditional use of these compounds has been reported [79-81]. Additionally, the effect of glycosides against osteoarthritis were reported, with representative glycosides described below.”

On page 12: “Polyphenols, which are compounds with a polyphenol structure, comprise two subgroups; flavonoids and non-flavonoids. There are three subgroups of non-flavonoids with different structures: phenolic acids, stilbenes, and lignans [94]. Phenolic acids are hydroxybenzoates that contain some phenolic rings and at least one carboxylic acid. Stilbenes commonly have a C6-C2-C6 structure; resveratrol is the most common stilbene. Lignans are low-molecular weight polyphenols that exist as phenolic dimers with a 2,3-dibenxylbutane structure.”

On page 13: “In addition to those mentioned above, many other natural compounds are found in plants and animals and have been used for drug discovery or bioassays because of their bioactivity. As natural compounds, secondary metabolites are used because they confer advantages but are not necessary for survival. Among the compounds not included in the previous category, some compounds reported as effective in osteoarthritis are introduced below.”

Comment:

[3] Following the comment 2, where is the section 2.2.3?

Author response and action:

Thank you for your comment. We have corrected the section number, which has been highlighted in the revised manuscript.

Comment:

[4] Figure 1, 2, 3, 5, 6, 8, 9, 10, 11, and 13 (for the chemical structures of the compounds) should be combined into one figure, schematically shown all the compounds.

Author response and action:

Thank you for your comment. We have placed the separate figures in one figure named Figure 1, which has been highlighted in the revised manuscript.

Comment:

[5] The image resolution of figure 4, 7, 12, and 14 is not high enough for publication.

Author response and action:

We agree with the reviewer’s comment. We have increased the image resolution of Figure 4, 7, 12, and 14 and changed Figures 4, 7, 12, and 14 to Figures 2, 3, 4, and 5, respectively.

Comment:

[6] Format issues. For example, ‘2.2.5.1.3`-. Sialyllactose’ (line 347).

Author response and action:

Thank you for your comment. We have corrected the manuscript to correct such formatting errors. The relevant sentences are highlighted in the revised manuscript.

2.2.4.1. 3'-Sialyllactose (line 510)

Reviewer 2 Report

In the article “Therapeutic Single Compounds for Osteoarthritis Treatment”, the authors have discussed the role of single compound for the treatment of OA. OA is a joint degenerative disease and there is no disease modifying drug available for its treatment. The idea of the review is good, however, the presentation is poor. Most of the places, the authors have just described the results of previous studies. The review lacks critical analysis of the data and limitations associated with individual polyphenols. The abstract which describes the feasibility of single compound in clinical trials is not justified in the main text. There are several studies on clinical trials of individual compounds which are not included in this review. There are some specific comments below.

Line 40-41: “Recent study--- (4)” the references cited here was published in 2008. Change it to “recent” study. Overall, the introduction is poor and does not justify the title.

The first paragraph in 2.1.1 on extract is not needed here.

Expand the abbreviations at first appearance.

The discussion of apigenin and other compounds on cancer or other diseases is not relevant to this review. Authors should focus only on OA.

There are several recent studies describing the effect of Butein, Wogonin, Morin, Quercetin etc.

Overall, the review is nothing more than describing the results of previous studies.  

Author Response

Reviewers' comments:

 Reviewer #2:

Comments to the Author
Major corrections:

Comment:

In the article “Therapeutic Single Compounds for Osteoarthritis Treatment”, the authors have discussed the role of single compound for the treatment of OA. OA is a joint degenerative disease and there is no disease modifying drug available for its treatment. The idea of the review is good, however, the presentation is poor. Most of the places, the authors have just described the results of previous studies. The review lacks critical analysis of the data and limitations associated with individual polyphenols. The abstract which describes the feasibility of single compound in clinical trials is not justified in the main text. There are several studies on clinical trials of individual compounds which are not included in this review. There are some specific comments below.

Author response and action:

Thank you for reviewing our manuscript. We have added a critical analysis of the data and limitations of individual compounds throughout the text.

2.2.1.1. Apigenin (page 4)

In an in vivo study, apigenin attenuated cartilage erosion, bone loss, and catabolic factors and reduced the expression of pro-inflammatory cytokines in a MIA-induced rat OA model (Daru 2020, 28, 443-453).

This study defined the inhibition mechanisms of apigenin against OA in vitro and in vivo; however, this study did not determine the upstream molecules of NF-κB and JNK signaling. HIF-2α is a key regulator in OA, and HIF-2α ex-pression induces the expression of catabolic factors that induce OA by degrading cartilage and inducing inflammation. Therefore, apigenin may be among the first drugs targeting HIF-2α against OA.

2.2.1.2. Cirsimarin and Cirsimaritin (page 5)

The in vivo effects of cirsimarin or cirsimaritin on OA have not been reported, although these compounds were shown to have anti-inflammatory activities. Cho et al. (2019) performed in vitro analysis and showed that cirsimarin and cirsimaritin decrease IL-1β-induced HIF-2α and COX2 expression but not MMP3 and MMP13 expression (J Cell Mol Med 2019, 23, 5369-5379)

2.2.1.3. Jaceosidin (page 6)

As only Yang et al. have reported the effects of jaceosidin on OA, further studies are needed to confirm this result.

2.2.1.4. Eupatilin (page 7)

However, Jeong et al. used only an MIA OA model, which is not suitable for evaluating the mild progression of OA. In addition, Lou et al. did not evaluate autophagy flux; there is limited information on whether eupatilin-induced autophagy has positive or negative effects on OA.

2.2.1.5. Genistein (page 8)

However, this study did not define the apoptosis mechanisms e.g., caspase-dependent or caspase-independent. When genistein was orally administrated in an anterior cruciate ligament transection (ACLT) rat model, the collagen and acid glycosaminoglycan content was increased compared to that in the control OA rat model, whereas the level of TNF-α and IL-1β decreased (Mol Med Rep 2020, 22, 2032-2042). Additionally, genistein showed anti-inflammatory effects in IL-1β-stimulated human OA chondrocytes (Nutrients 2019, 11). Genistein attenuated NOS2, COX2, and MMP expression in IL-1β-stimulated chondrocytes through activated nuclear factor erythroid 2-related factor 2 (NRF2)/heme oxygenase-1 (HO-1) signaling.

2.2.1.6. Epigallocatechin Gallate (EGCG) (page 11)

EGCG can also inhibit the mRNA and protein expression of MMP-1, MMP-13, and COX-2 in human or mouse chondrocytes under IL-1β stimulation, thereby inhibiting GAG release from cartilage tissues (J Pharmacol Exp Ther 2004, 308, 767-773; Eur J Nutr 2018, 57, 917-928).

Furthermore, numerous studies suggested that EGCG can be used as a dietary supplement for arthritis (J Agric Food Chem 2019, 67, 6476-6486; J Sci Food Agric 2018, 98, 1653-1659).

2.2.2.1. Paeoniflorin (page 9)

Further, paeoniflorin suppressed the migration of fibroblast-like synoviocytes from patients with RA or OA by blocking CXCR4-Gβγ-PI3K/AKT signaling (Biochemical and Biophysical Research Communications 2020, 526, 805-812).

Another study reported that paeoniflorin blocked chondrocyte apoptosis induced by IL-1β by downregulating Bcl2 and Bcl-2-associated X-protein levels as well as caspase 3 activity.

The mechanisms of paeoniflorin on OA have only been demonstrated in vitro; therefore, in vivo animal studies are needed to clarify the mechanisms of the effects of paeoniflorin.

2.2.2.2 Clematis Saponins (page 10)

As inhibition of chondrocyte apoptosis is the main mechanism of clematis saponins against OA, other mechanisms should be investigated for drug development.

2.2.3.1. Resveratrol (Veratrum grandiflorum) (page 10)

This study did not show histology images, although animal experiments were performed. The anti-osteoarthritic effects of resveratrol via NF‑κB signaling were recently reported. For example, resveratrol regulates NF‑κB signaling in IL-1β-induced chondrocyte injury (J Orthop Surg Res 2020, 15, 424). The anti-inflammatory effects of resveratrol are also involved in NF‑κB signaling (Drug Des Devel Ther 2020, 14, 2079-2090; Exp Ther Med 2020, 19, 2343-2352)

Only the in vitro mechanism was determined, which is a limitation of this study. Importantly, intra-articular injection of resveratrol exerted a curative effect by preventing inflammation and cartilage destruction (J Oral Maxillofac Surg 2020, 10.1016).

2.2.3.2. Curcumin (Curcuma longa L.) (page 11)

In addition, topically applied curcumin nanoparticles were localized within the infrapatellar fat pad and effectively reduced the expression of pro-inflammatory mediators (Arthritis Research & Therapy 2016, 18, 128-128). This study provides the first evidence that single compounds can reach the articular cartilage area and exert therapeutic effects.

Additionally, it has been reported that dietary supplementation or ointment application reduced osteoarthritic pain in a mouse model and clinical trials (Animals (Basel) 2020, 10; BMC Complement Med Ther 2020, 20, 305).

2.2.5.1. 3'-Sialyllactose (page 12)

Studies of the effects of 3'-sialyllactose against rheumatoid arthritis or OA rheumatoid arthritis have been performed using suitable in vitro and in vivo mouse models, although other carbohydrate-type compounds must be evaluated.

2.2.5.2. Triptolide (Tripterygium wilfordii Hook. f.) (page 13)

The exact molecular mechanisms of the effects of triptolide on OA remain unclear.

2.2.5.3. Hyaluronic Acid (HA) (page 14)

Recently, to increase the efficacy of pain relief by HA, combination approaches have been attempted using a PPAR-δ agonist or mesenchymal stem cells. (Regen Ther 2020, 15, 103-111; Med Arch 2020, 74, 387-390)

Comment:

Line 40-41: “Recent study--- (4)” the references cited here was published in 2008. Change it to “recent” study.

Author response and action:

We have replaced these contents with a recent reference as shown below.

According to the Center for Disease Control and Prevention (CDC) estimates, there are 78 million US adults will have arthritis in the year 2040 (Morbidity and Mortality Weekly Report 2017, 66, 246-253).

Comment:

Overall, the introduction is poor and does not justify the title.

Author response and action:

We have edited the entire Introduction section to correspond with the manuscript title.

Comment:

The first paragraph in 2.1.1 on extract is not needed here.

Author response and action:

Thank you for your comment. We feel that the single compounds from natural plants are the main focus of this manuscript. The functions of total extracts of compounds are also important in studies of natural or herbal plants. Moreover section 2.2 describing single compounds studies follows section 2.1. The part describing the extract in section 2.1 will clarify these points for the reader.

Comment:

Expand the abbreviations at first appearance.

Author response and action:

We have checked all abbreviations in the text and have defined them at the first mention.

Comment:

The discussion of apigenin and other compounds on cancer or other diseases is not relevant to this review. Authors should focus only on OA.

Author response and action:

Based on the reviewer’s comment, we have deleted the description of cancer and other diseases from the text.

Comment:

There are several recent studies describing the effect of Butein, Wogonin, Morin, Quercetin etc.

Author response and action:

Based on the reviewer’s comment, we have added information on butein, wogonin, morin, and quercetin to the text. (pages 11–13)

Comment:

Overall, the review is nothing more than describing the results of previous studies.

Author response and action:

We have added additional analysis data and discussed the limitations of individual compounds evaluated in previous studies. We have described previous studies on butein, wogonin, morin, and quercetin in the text,including clinical trials of these compounds.

Round 2

Reviewer 1 Report

The text and the chemical structures are overlapped in figure 5.

Author Response

Comment:

[1] The text and the chemical structures are overlapped in figure 5.

Author response and action:

We agree with the reviewer’s comment. The text and the chemical structures are overlapped in figure 5.

Reviewer 2 Report

The authors have provided satisfactory answer and have addressed the previous comments. I have no further concerns.